



# Hydrometeorological dataset of West Siberian boreal peatland: a 10-year record from the Mukhrino field station

Egor Dyukarev[1,2], Nina Filippova[1], Dmitriy Karpov[1], Nikolay Shnyrev[3], Evgeny Zarov[1], Ilya Filippov[1], Nadezhda Voropay[2,4], Vitaly Avilov[5], Arseniy Artamonov[6], Elena Lapshina[1]

[1]Yugra State University, Khanty-Mansiysk, 628012, Russia
[2]Institute of Monitoring of Climatic and Ecological System SB RAS, Tomsk, 634055, Russia
[3]Faculty of Soil Science, Lomonosov Moscow State University, Moscow, 119991, Russia
[4]V. B. Sochava Institute of Geography SB RAS, Irkutsk, 664033, Russia
[5]A. N. Severtsov Institute of Ecology and Evolution RAS, Moscow, 119071, Russia
[6]A. M. Obukhov Institute of Atmospheric Physics, Moscow, 119017, Russia

*Correspondence to*: Egor Dyukarev (egor@imces.ru)

## Abstract

Northern peatlands represent one of the largest carbon pools in the biosphere but the carbon they store is increasingly vulnerable to perturbations from climate and land-use change. Meteorological observations taken directly at peatland areas in

Siberia are unique and rare, while peatlands are characterized by a specific local climate. This paper presents a hydrological and meteorological dataset collected at the Mukhrino peatland, Khanty-Mansi Autonomous Okrug – Yugra, Russia, over the period of 8 May 2010 to 31 December 2019. Hydrometeorological data were collected from stations located at a small pine-shrub-*Sphagnum* ridge and *Scheuchzeria-Sphagnum* hollow at ridge–hollow complexes of ombrotrophic peatland. The monitored meteorological variables include air temperature, air humidity, atmospheric pressure, wind speed and direction,

incoming and reflected photosynthetically active radiation, net radiation, soil heat flux, precipitation (rain) and snow depth. A gap-filling procedure based on the Gaussian process regression model with an exponential kernel was developed to obtain continuous time series. For the record from 2010 to 2019, the average mean annual air temperature at the site was −1.0 °C, with the mean monthly temperature of the warmest month (July) recorded as 17.4 °C and for the coldest month (January) −21.5 °C. The average net radiation was about 35.0 W m$^{-2}$, and the soil heat flux was 2.4 and 1.2 W m$^{-2}$ for the hollow and the

ridge sites, respectively.

The presented data are freely available through Zenodo (https://zenodo.org/record/4323024), last access: 15 December 2020) and can be used in coordination with other hydrological and meteorological datasets to examine the spatio-temporal effects of meteorological conditions on local hydrological responses across cold regions.

## 1 Introduction

The availability of hydrometeorological data is limited in northern latitudes because of a sparse monitoring network, harsh weather, and the high cost of experiments and instrument maintenance in these environments (Rasouli et al., 2019). The number



of stations that record a complete hydrometeorological dataset in the northern latitudes is limited and declining (Laudon et al., 2017). Weather stations located directly at peatland areas are unique and rare, while peatlands are characterized by a specific local climate (Worrall et al., 2019; Kiselev et al., 2019; Koronatova et al., 2018).

Northern peatlands developed mostly after the last deglaciation in the circum-Arctic region and represent one of the largest carbon pools in the biosphere (Yu, 2012). Peatlands clearly play a significant role in global carbon cycling and the carbon they store is increasingly vulnerable to perturbations from climate and land-use change (Amesbury et al., 2019). Temperature is the most important long-term driver of peat accumulation in northern peatlands, and excessive moisture is deemed a necessary condition for peatland development, maintenance and C preservation (Loisel et al., 2020). Increased global warming, such as

the increased temperature and resulting water table drawdown, may imbalance peatland carbon cycles, resulting in a large feedback to the global climate (Packalen et al., 2016; Samson et al., 2018; Dyukarev et al., 2019).

Large peatland systems in Western Siberia occupy about 28% of the area (Terent'eva et al., 2017) and continued observations of atmospheric conditions and upper soil layers are therefore of great importance. Only a few modern hydrometeorological datasets are available for the Northern part of Russia (i.e. Beer et al., 2013; Heimann et al., 2014; Boike et al., 2019) and they

are primarily related to Arctic sites. Hydrometeorological data are required for the study of the ecosystem–atmosphere exchange (Alekseychik et al., 2017; Holl et al., 2019), biochemical processes in peat (Szajdak et al., 2016; Djukic et al., 2018), hydrology (Bleuten et al., 2020), and microbiology including mycology (Filippova and Lapshina, 2019).

Mukhrino Field Station (MFS - mukhrinostation.com) was established in 2009 as part of the UNESCO chaired Environmental Dynamics and Global Climate Change (EDCC) of the Yugra State University (Khanty-Mansiysk, Russia). It is equipped with

modern facilities allowing the conduct of year-round long-term scientific research, scientific excursions, workshops, symposia and other events at the national and international level.

MFS became part of INTERACT– International Network for Terrestrial Research and Monitoring in the Arctic (https://eu-interact.org/field-sites/mukhrino-field-station/) in 2012 and has developed its infrastructure in line with the Network activities. The station is regularly visited by international research groups through the INTERACT Transnational Access (TA) working

package to conduct field work at the station. About 30 international research projects were conducted at MFS through INTERACT TA during the last 5 years. The participation in INTERACT is propelling the development of other functions of Arctic Stations, e.g. monitoring standardization, data management, communication, safety regulations, science outreach and other activities, thus developing the station according to international best practices.

## 2 Site description

MFS is located in the central part of Western Siberia in the Middle Taiga biogeographic zone, 30 km to the south-west of Khanty-Mansiysk, on the left upper terrace of Irtysh River (near the confluence with Ob River) at the Mukhrino peatland (Fig. 1). The wide area to the south-west is represented by the paludified peatlands and lakes landscape of the Kondinskaya Nyzmennost interspersed by forests along the rivers. The Mukhrino peatland is located at the northernmost part of it, bordering with the Ob River floodplain and distinguished from the other surroundings by an oval shape about 10 x 5 km in size. The

MFS research polygon is located in the north-east part of the peatland and covers an area of about 1 km$^2$, providing a system of walking boards with a total length of about 2 km, an energy supply complex and permanent monitoring plots for peatland ecosystem studies with hydrometeorological equipment.

Different aspects of Mukhrino peatland were described in a series of publications over the history of MFS (Lapshina et al., 2015). To mention a few, the hydrological dynamics and fire history for the last millennium were described in (Lamentowicz
et al., 2015; Lamentowicz et al., 2016). The hydrological model of a peatland is provided by Bleuten et al. (2020). And the net ecosystem exchange is described by (Alekseychik et al., 2017; Dyukarev et al., 2019). The present stage of the peatland development is represented by a raised oligotrophic bog with a mosaic of ridge-hollows, oligo-mesotrophic fens and treed bogs micro-landscapes. A few secondary lakes up to 300 m in width are located in the most waterlogged areas and the central part of the peatland is occupied by a wide watercourse. The average peat depth is 3.3 m, with the longest core depth (located
at an ancient alley) reaching about 5 m (Bleuten et al., 2020). The most abundant peat type is Sphagnum peat, with pH 3.5–5 and electric conductivity from 0 to 200 µSm m$^{-2}$ (Sabrekov et al., 2011).

The vegetation comprises oligotrophic communities dominated by various *Sphagnum* species. The highest levels with the ground water below 30 cm (about a third of the peatland area) are covered by pine-dwarf shrub-*Sphagnum* communities (so-called "ryams"), dominated by *Pinus sylvestris* and *P. sibirica* and several dwarf shrubs (*Chamaedaphne calyculata*, *Ledum*
*palustre*, *Andromeda polifolia*, *Vaccinum uliginosum*, *V. oxycocci*). Herb species, such as *Rubus chamaemorus*, *Carex globularis*, *Eriophorum vaginatum* and *Drosera* spp. are scarce in diversity and density. The dominating species of *Sphagnum* here is *S. fuscum*, with other species (*S. magellanicum*, *S. angustifolium*, *S. capillifolium*) in admixture. The pine-dwarf shrub-Sphagnum communities also participate in ridge–hollow complexes and their variations, the most abundant landscapes of Mukhrino peatland, with minor differences in plant composition. The lower positions of the landscape with ground water level
0–15 cm are covered by graminoid-*Sphagnum* communities. The dwarf shrubs are represented by *Andromeda polifolia* and *Oxycoccus palustris*. The herbs include several species: *Scheuchzeria palustris*, *Carex limosa*, *Eriophorum russeolum*, *Drosera* spp. Several hydrophilic *Sphagnum* species are dominant in the moss cover: *S. balticum, S. papillosum, S. jensenii, S. majus, S. lindbergii.* The most waterlogged conditions along peatland watercourses contain sparse vegetation from several floating species like *Menyanthes trifoliata* and *Sphagnum majus*.

The hydrometeorological complex described in the paper is located inside the irregular ridge–hollow complex with nearly equal proportions of the pine-dwarf shrub-Sphagnum ridges and graminoid-*Sphagnum* hollows. The trees' mean height at the ridges is about 3 m, with sparse trees reaching up to 10 m height. Waterlogged areas, lakes or streams with open water are absent in the near vicinity (but exist at about 500 m distance).

## 3 Data description

Hydrometeorological data are available for MFS from 2010 to 2019 for two sites at boreal raised peatland in typical microlandscape forms. Data on air temperature, air humidity, atmospheric pressure, wind speed and direction, incoming and



outgoing shortwave radiation, net radiation, and soil heat flux were recorded at three automated weather stations. Two stations were located at a small pine-shrub-*Sphagnum* ridge and one station at a *Scheuchzeria-Sphagnum* hollow. The hollow represents a wet-site with the water level near the surface (0–15 cm), and the ridge is a relatively dry site with the water level at 20–40

cm depth. All sensors were connected to four Campbell Scientific data loggers CR10X with AM16/32A multiplexers to collect data. The data were collected at scan rate of 30 seconds and averaged for 15/30/60 min intervals by data logger software. Table 1 lists the sensors measuring the meteorology parameters and their location. The weather stations were assembled and tested by IN SITU INSTRUMENTS AB (Sweden).

An air temperature and humidity probe was covered by a naturally ventilated radiation shield ROTRONIC AC1000. An

atmospheric pressure sensor was mounted inside the enclosure case for the data logger. Wind speed and direction sensors were installed on a 10 m mast at the ridge site and a 2 m tripod at the hollow site. The distance between the mast and tripod is about 15 m. Net radiation, upward and downward photosynthetically active radiation (PAR) sensors were mounted at each site on a 2 m support crossarm CM200 with a levelling fixture. Two soil heat flux sensors were installed at the ridge site to cover the spatial variability of fluxes due to the inhomogeneous microlandscape. The soil heat flux sensors were initially installed within

the moss layer at a depth of 10 cm and checked in 2015, and moved again to a 10 cm position. In 2020 the heat flux sensors were found at a depth of 20 cm due to the growth of mosses and increase of the dead moss layer thickness. A self-calibrating procedure was applied every 3 minutes each hour for calibration of the soil heat flux sensors. The heat flux values generated during the self-calibration process were excluded from time-averaging. Liquid precipitation was measured by an unshielded tipping bucket rain gauge deployed at the surface level after the disappearance of the snow cover. The automated measurements

in 2010-2013 were accompanied by routine manual meteorological observations of air temperature and humidity, and precipitations. Measurements of the snow depth were made daily from November to April in 2011–2014 using permanently installed snow depth lines with a 1 cm scale. Snow precipitation was measured manually using a rain gauge of Tretyakov construction, and the frequency of measurements varied from a day to a week during the whole period of observations.

Some other characteristics were recorded by the weather stations, such as the standard error of the wind direction, average soil

temperature in the 0–20 cm layer and battery voltage, but are not discussed here. Soil temperature measurements at five depths down to 50 cm were made using a Hukseflux Thermal sensor STP01 at four sites (two at ridges and two at hollows). Soil surface temperature was measured using averaging soil thermocouple probe TCAV (Campbell Sci. Inc.) Soil temperatures data are not presented here due to the high level of noisy disturbed data.

## 3.1 Data processing

All time series data were collected and stored in the data loggers at the weather stations. Several times a year data were manually downloaded, rearranged and archived at the Yurga State University. Before 2020 raw data were publicly available through a shared Google Drive directory (Mukhrino Weather Station, 2020; Mukhrino Field Station, 2020). Due to a power system malfunction, the weather stations operated in 2010–2012 for less than half the year. There were data missing each year





and the number of full days with available data is shown in Fig. 2. The weather stations recorded data at 15 min intervals in
2010–2011, hourly intervals in 2012–2013, and half-hourly since 2014, but are reported hourly in this paper. The missing data
were denoted by "NA". Raw data were thoroughly checked for errors and erroneous data were removed. Soil heat flux sensors
produce unnatural spikes in measurement data so these spikes were removed from the data. If the deviation from the moving
average is greater than 5 standard deviations for a centered time window of three days, the data point is discarded. The rejected
data were denoted by "NA".

Factory calibration coefficients were applied to the data of the PAR and net radiation sensors. All the other sensors' calibration
coefficients were implemented in the data recording software in the data logger. Surface albedo was calculated as the ratio of
incoming and reflected PAR for values of incoming PAR exceeding 30 µmol m$^{-2}$ s$^{-1}$.

**3.2 Data gap-filling**

The number of missing observation data in the early period of automated station operation is high. Therefore, different methods
for the gap-filling procedure were tested. Continuous weather data on various meteorological characteristics are required to
produce a continuous gap-free data set.

The fourth-generation reanalysis ERA5 was chosen as a source of continuous meteorological data. The ERA5 dataset showed
the best performance with NASA's most recent satellite-based dataset (Hennermann, 2019). ERA5 updates ERA-Interim using
the most recent ECMWF model (Hersbach et al., 2018), adopting a four-dimensional variational data assimilation system (4D-
VAR). It improves the correction of satellite observations and ground-based radar (Beck et al., 2019). Hourly data for 46
meteorological parameters (see Table 2) were provided by the ECMWF downloaded from the Climate Data Store (Muñoz-
Sabater et al., 2019) for the period from January 2010 to December 2019.

The dataset has a spatial resolution of 0.1°×0.1°, which approximately corresponds to 11.1 km in latitude and 5.4 km in
longitude for the MFS area. The ERA5-Land time series for a grid point with coordinates 60.9° N 68.7° E was used as reference
continuous meteorological data. It is clear that direct comparison of local observation data with the global reanalysis product
is senseless, because the data sets have completely different origins and purposes. Nonetheless, ERA5-Land reanalyses
reproduce local weather conditions with reasonable accuracy. The differences in the time series of observed and reanalysis
data are high, but the linear correlation is good (Berg et al., 2018; Kharyutkina et al., 2019).

Several regression models were tested for the gap-filling procedure. The model performance was estimated using the root
mean squared error (RMSE) value. Model parameters were estimated on the training set and its performance was assessed
with the validation set. The model used for validation is based on 75% of the data. The final model is trained using the full
data set. Regression models were optimized using the Regression Learner Toolbox from MATLAB. The Regression Learner
performs supervised machine learning by supplying a known set of observations of input data (predictors) and known
responses. The list of tested regression models includes: linear, interactions linear, robust linear, stepwise linear, quadratic,
fine/medium/coarse tree, support vector machine regression with linear, quadratic, cubic and Gaussian kernel, Gaussian
process regression with rational quadratic, squared exponential, Matern 5/2 and exponential kernel (The MathWorks, 2019).



It was found that the Gaussian process regression model exponential kernel gives the minimal RMSE for all observation time series.

Wind direction and wind speed observation data were recalculated into meridional and zonal (U and V) wind components.

Relative air humidity was recalculated into water vapor pressure. Before model training, missing snow depth data when ERA5 reanalysis indicated the absence of snow cover was set to zero. Missing incoming/reflected PAR data was set to zero for the night time. Night time periods were determined as the time when downwards solar radiation from ERA5 reanalysis is zero.

Models were trained for 19 time series of meteorological parameters, including air temperature, air absolute humidity, incoming and reflected PAR, net radiation, U and V wind components for both the ridge and hollow sites. Regression models

for atmospheric pressure and snow depth were trained using observation data for a single site. Three models were trained against soil heat flux observation data at the ridge (two sites) and the hollow (one site).

All 46 parameters from the ERA5-Land reanalysis were used as input variables for the models. The model data have an extremely high linear correlation coefficient (r>0.99) with the observation data. Long-term mean, range and errors for the model data are shown in Table 3. Comparisons of the observed and modeled time series, residuals and probability distributions

are given in the Supplementary Material Figs. S1 to S19. Mean, mean absolute and root mean squared errors for all the modeled time series are small (Table 3) and therefore the model data can be used to interpolate the observations into the gaps. The probability distributions of the observed and model data (Figs. S1–S19) are very close. Extremely small errors for model data were obtained due to a large amount of input discontinuous variables (46) from ERA5-Land reanalysis.

Gap-filled time series were constructed from all the available data of observations, replacing the missing data with model

values. Negative model data for incoming and reflected PAR were set to zero. Filling the gaps in data of the precipitation time series is a very complex task and was not solved in the present research. Comparison of the liquid precipitation data with total precipitation from ERA5-Land reanalyses is shown in Fig. S20 in the Supplementary Material. Fig. S21 illustrates the calculated albedo variations.

## 4 Data examples

Figures 3–6 illustrate the annual, seasonal and diurnal variations of the hydrometeorological parameters observed at MFS. The monthly air temperature varies from 13.8 to 17.4 °C in July and from –27.8 to –17.3 °C in January (Fig. 3a), whereas the absolute temperature minimum was –45.0 °C at 22:00 on 21 December 2016, and the absolute temperature maximum was 32.9 °C at 16:00 on 5 August 2016. The average, minimal and maximal values for all the observed variables are shown in Table 3. The air humidity in winter is much lower than in summer. The monthly water vapor pressure varies from 0.05 to 0.16 kPa in

January and from 1.22 to 1.69 in July (Fig. 3b). The differences between measurements of air parameters obtained at the ridge and the hollow sites are insignificant. The two sites are closely situated and intense air mixing equalizes the air conditions.

The incoming PAR registered at both sites (Fig. 4a) has a maximum at noon, and the value of the maximum rises from December to July. The amount of reflected PAR is closely related with the state of the surface. The albedo for the PAR range



(the ratio of reflected and incoming PAR) in summer is about 0.03 and 0.06 at the hollow and ridge sites, respectively.
Extremely small albedo values are related with the spectral range of the PAR sensor. The PAR range albedo can be useful for characterizing the vegetation greenness. The albedo in winter at snow-covered surfaces is about 0.95 at the hollow site and 0.8 at the ridge site, where small dark branches of trees are present. The net radiation balance has close maximal values at both sites (Fig. 4b), but the diurnal course of net radiation at the ridge is shifted one hour later compared with the hollow site. The January net radiation is negative and varies within a range from –8 to –18 W m$^{-2}$ during a day. The daily averaged soil heat
flux is negative from October to March. The maximal heat flux into the soil was observed in June at approximately 18:00 local time. The amplitude of diurnal variations of soil heat flux at the hollow is 2–3 times higher than at the ridge. The soil heat flux sensors at the ridge were located under the porous mat of weakly decomposed dead mosses isolating the peat layers from heating.

The snow cover onset date varies from 9 October in 2014 to 3 November in 2010 (Fig. 5). Maximal snow storages were
recorded on 18 March 2013 when the snow depth reached 95 cm. The winter of 2010–2011 was the season with the weakest snow pack. The snow cover at the end of winter on 16 February 2011 was only 64 cm. Complete melting of the snow can take place between 16 April and 19 May depending on the year. The average duration of the snow cover period is 191 days. South-south-east winds prevail at the observation site (Fig. 6), but winds with speeds above 5 m/s are mostly of north-east origin. The median wind speed value at 10 m is 1.8 m/s, while at 2 m above the surface it is only 1.0 m/s. The wind rose structure is
similar for all observation years, except 2017 and 2019.

The automated weather station at the MFS was rebuilt in October 2020. All the sensors were connected to a new data logger (CR1000X) through four multiplexers. A four-channel net radiometer CNR1 (Kipp&Zonnen) was installed for measuring the incoming short-wave, incoming long-wave, surface-reflected short-wave and outgoing long-wave radiation. A new rain-gauge MPDO-500.120 Volna (MeraPribor) with heater will allow winter and summer precipitation to be registered. We will continue
to update these data sets for use in baseline studies, as well as to assist in identifying important processes and parameters through conceptual or numerical modeling.

## 6 Data availability

The database presented and described in this article is available for download from Zenodo https://zenodo.org/record/4323024 (Dyukarev et al., 2020). Gap-filled, quality controlled, and raw observation data are provided in separate files in csv format.

**Author contributions.** ED, NF, NV, EZ, EL cleaned, organized, and corrected the data and wrote the first draft of the paper. ED and NV developed the gap-filling procedure. NS, DK, IF, AA and VA designed and built the instrumental stations, collected data, managed the data collection over the last decade, and contributed to the writing of the manuscript.

**Competing interests.** The authors declare that they have no conflict of interest.



**Disclaimer**. Any reference to specific equipment types or manufacturers is for informational purposes and does not represent a product endorsement.

**Acknowledgements**. The research was carried out within the grant of the Tyumen region Government in accordance with the Program of the World-Class West Siberian Interregional Scientific and Educational Center (National Project "Nauka"), Russian Fund for Basic Researches in the framework of scientific projects 15-44-00091, 18-05-00306, 18-44-860017, and under support the Yugra State University grant 17-02-07/58 from 14.02.2020. The Mukhrino Field Station infrastructure development was supported by INTERACT project - International Network for Terrestrial Research and Monitoring in the
Arctic (grant nimbe0sr: 730938, 871120). Field work support by MFS staff, Yaroslav Solomin and Alexey Dmitrichenko was essential in accurate data collection in adverse conditions.

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





**Tables and figures**

**Table 1.** List of sensors, parameters and installation site at Mukhrino field station, 2010-2019.

| n | Parameter | Equipment | Ridge | Hollow |
|---|---|---|---|---|
| 1 | Air temperature and humidity at 2 m | Rotronic HC2A-S3 | 1 | 1 |
| 2 | Atmospheric pressure | Campbell Scientific CS105 PTB101b | 1 | - |
| 3 | Wind speed and direction at 2 m | Young Wind Monitor 05103 | - | 1 |
| 4 | Wind speed and direction at 10 m | Young Wind Monitor 05103 | 1 | - |
| 5 | Incoming PAR | Li-Cor LI-190R | 1 | 1 |
| 6 | Reflected PAR | Li-Cor LI-190R | 1 | 1 |
| 7 | Net radiation balance | Kipp & Zonen NRLite | 1 | 1 |
| 8 | Ground heat flux | Hukseflux Heat Flux Sensor HFP01SC | 2 | 1 |
| 9 | Precipitation (summer) | HOBO Data Logging Rain Gauge RG3-M | - | 1 |
| 10 | Snow depth | Manual observations | - | 1 |
| 11 | Surface albedo | Calculated from Li-Cor LI-190R | 1 | 1 |

**Table 2.** List of meteorological variables from ERA5-Land reanalysis used for gap-filling, minimal, maximal and average values for 2010-2019. mwe – meter of water equivalent.

| n | Variable, unit | n | Variable, unit |
|---|---|---|---|
| 1 | 2 m temperature, °C | 24 | Skin reservoir content, mwe |
| 2 | Skin temperature, °C | 25 | Runoff, m s$^{-1}$ |
| 3 | 2 m dewpoint temperature, °C | 26 | Surface runoff, m s$^{-1}$ |
| 4 | Relative air humidity, % | 27 | Sub-surface runoff, m s$^{-1}$ |
| 5 | 10 m U wind component, m s$^{-1}$ | 28 | Snow cover, % |
| 6 | 10 m V wind component, m s$^{-1}$ | 29 | Snow depth, m |
| 7 | Wind speed at 10 m, m s$^{-1}$ | 30 | Snow depth water equivalent, mwe |
| 8 | Wind direction at 10 m, deg | 31 | Snow albedo |
| 9 | Surface pressure, hPa | 32 | Snow density, kg m$^{-3}$ |
| 10 | Total precipitation, mm h$^{-1}$ | 33 | Temperature of snow layer, °C |
| 11 | Surface solar radiation downwards, W m$^{-2}$ | 34 | Snowfall, mwe s$^{-1}$ |
| 12 | Surface thermal radiation downwards, W m$^{-2}$ | 35 | Snowmelt, mwe s$^{-1}$ |
| 13 | Surface net solar radiation, W m$^{-2}$ | 36 | Snow evaporation, mwe s$^{-1}$ |
| 14 | Surface net thermal radiation, W m$^{-2}$ | 37 | Leaf area index (LAI), high vegetation, m$^2$ m$^{-2}$ |
| 15 | Forecast albedo | 38 | LAI, low vegetation, m$^2$ m$^{-2}$ |




| 16 | Surface sensible heat flux, W m$^{-2}$ | 39 | Soil temperature (ST) level 1, °C |
| 17 | Surface latent heat flux, W m$^{-2}$ | 40 | ST level 2, °C |
| 18 | Potential evaporation, mwe s$^{-1}$ | 41 | ST level 3, °C |
| 19 | Evaporation (EV), mwe s$^{-1}$ | 42 | ST level 4, °C |
| 20 | EV from bare soil, mwe s$^{-1}$ | 43 | Volumetric soil water (VSW) layer 1, % |
| 21 | EV from open water surfaces, mwe s$^{-1}$ | 44 | VSW layer 2, % |
| 22 | EV from the top of canopy, mwe s$^{-1}$ | 45 | VSW layer 3, % |
| 23 | EV from vegetation transpiration, mwe s$^{-1}$ | 46 | VSW layer 4, % |


**Table 3.** Average, minimal, maximal and errors for model data for 2010-2019. ME – mean error, MAE – mean absolute error, RMSE – root mean squared error.

| n | id | Variable | Unit | Min | Max | Mean | ME | MAE | RMSE |
|---|---|---|---|---|---|---|---|---|---|
| 1 | taH | Air temperature Hollow | °C | -45.01 | 32.76 | -1.05 | -2.09E-04 | 2.83E-01 | 2.09E-03 |
| 2 | taR | Air temperature Ridge | °C | -43.95 | 32.86 | -0.98 | -8.89E-04 | 2.05E-01 | 1.61E-03 |
| 3 | vpH | Vapor pressure Hollow | kPa | 0 | 2.70 | 0.64 | -1.14E-06 | 5.61E-04 | 3.92E-06 |
| 4 | vpR' | Vapor pressure Ridge | kPa | 0 | 2.63 | 0.63 | 3.27E-06 | 4.89E-04 | 3.69E-06 |
| 5 | iparH | Incoming PAR Hollow | μmol m$^{-2}$ s$^{-1}$ | 0 | 1520.8 | 190.2 | 3.06E-03 | 3.11E-01 | 1.16E-04 |
| 6 | iparR | Incoming PAR Ridge | μmol m$^{-2}$ s$^{-1}$ | 0 | 1587.3 | 192.7 | 9.56E-03 | 2.94E-01 | 1.22E-04 |
| 7 | rparH | Reflected PAR Hollow | μmol m$^{-2}$ s$^{-1}$ | 0 | 1283.0 | 49.7 | 2.25E-02 | 1.27E+00 | 8.01E-04 |
| 8 | rparR | Reflected PAR Ridge | μmol m$^{-2}$ s$^{-1}$ | 0 | 1087.5 | 36.7 | 1.14E-02 | 9.26E-01 | 6.94E-04 |
| 9 | nrH | Net radiation Hollow | W m$^{-2}$ | -166.3 | 668.6 | 34.6 | 9.52E-05 | 1.12E-01 | 9.10E-05 |
| 10 | nrR | Net radiation Ridge | W m$^{-2}$ | -203.6 | 661.5 | 35.3 | 1.42E-04 | 1.31E-01 | 1.14E-04 |
| 12 | shfH | Soil heat flux Hollow | W m$^{-2}$ | -24.0 | 30.7 | 2.4 | 1.25E-04 | 5.80E-03 | 4.24E-05 |
| 11 | shfR1 | Soil heat flux Ridge 1 | W m$^{-2}$ | -72.6 | 108.6 | 1.5 | -7.41E-04 | 1.91E-02 | 1.24E-04 |
| 13 | shfR2 | Soil heat flux Ridge 2 | W m$^{-2}$ | -33.7 | 32.7 | 0.9 | -3.27E-04 | 6.95E-03 | 4.88E-05 |
| 14 | w10U | U wind at 10 m | m s$^{-1}$ | -10.0 | 14.1 | 0.5 | -6.23E-06 | 8.03E-02 | 5.96E-04 |
| 15 | w10V | V wind at 10 m | m s$^{-1}$ | -11.6 | 11.9 | 0.1 | -2.22E-05 | 8.03E-03 | 6.24E-05 |
| 16 | w2U | U wind at 2 m | m s$^{-1}$ | -7.1 | 9.7 | 0.2 | -2.75E-06 | 1.72E-03 | 1.16E-05 |
| 17 | w2V | V wind at 2 m | m s$^{-1}$ | -5.6 | 8.1 | 0.2 | -3.98E-05 | 2.49E-02 | 1.76E-04 |
| 18 | prs | Atmospheric pressure | kPa | 98.1 | 108.0 | 103.2 | 5.85E-05 | 1.30E-03 | 8.29E-06 |
| 19 | sdp | Snow depth | cm | 0 | 95.0 | 25.4 | 6.15E-04 | 6.25E-03 | 3.62E-04 |

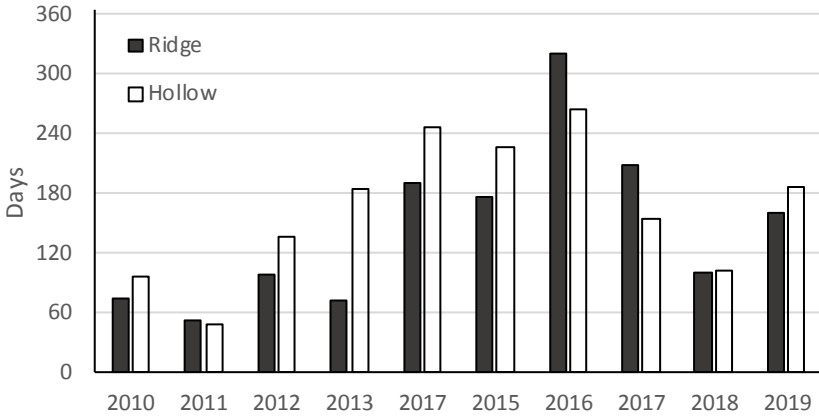

**Figure 1. Position of the weather stations near Khanty-Mansiysk and among bog–forest landscapes (top panels © Landsat 7 ETM+ image, NASA, 2020; © Quickbird image, Google Maps, 2020). Automatic weather stations located at the treed ridge and the *Sphagnum* hollow (bottom panels Foto: Nina Filippova).**

**Figure 2.  Number of days with hydrometeorological data at Mukhrino field station, 2010-2019.**

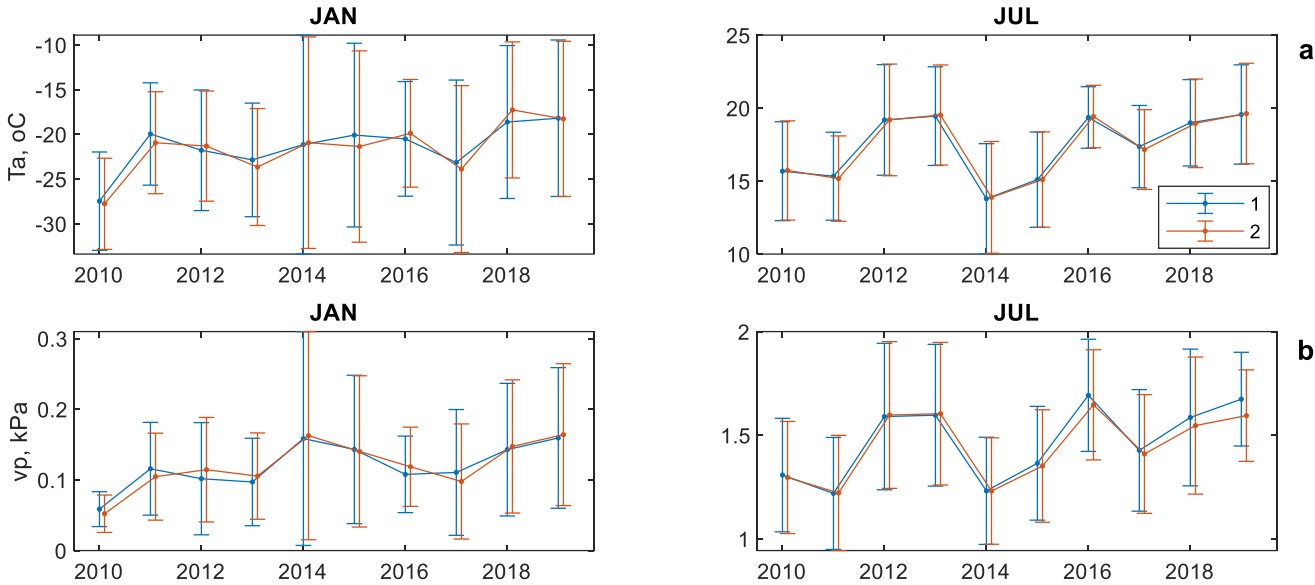


**Figure 3. Monthly averaged air temperature (a) and water vapor pressure (b) at the hollow (1) and the ridge (2) in January and July. Bars show standard deviations for a daily data.**

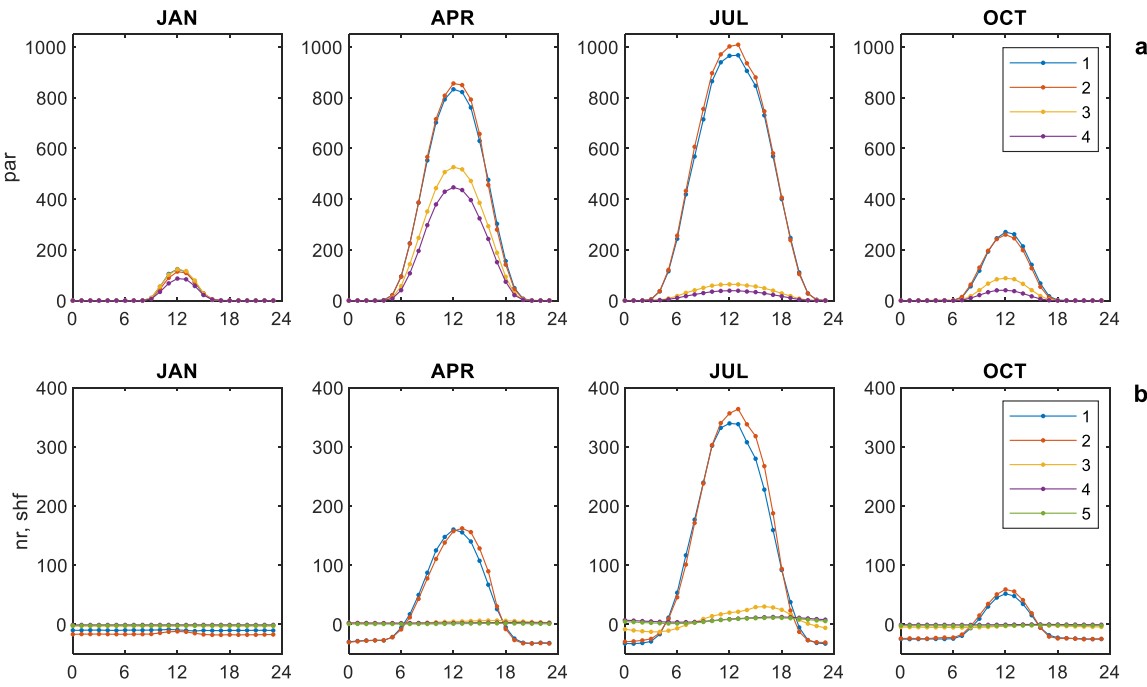


**Figure 4. Monthly averaged diurnal course of incoming and reflected photosynthetically active radiation (a) (µmol m$^{-2}$ s$^{-1}$) and net radiation balance, soil heat flux (b) (W m$^{-2}$) at hollow and ridge in January, April, July, and October. Mean values for 2010–2019. Legend: a) 1 – incoming PAR, hollow; 2 – incoming PAR, ridge; 3 – reflected PAR, hollow; 4 – reflected PAR, ridge. b) 1 – net radiation, hollow; 2 – net radiation, ridge; 3 – soil heat flux, hollow; 4 – soil heat flux, ridge, site 1; 5 – soil heat flux, ridge, site 2.**



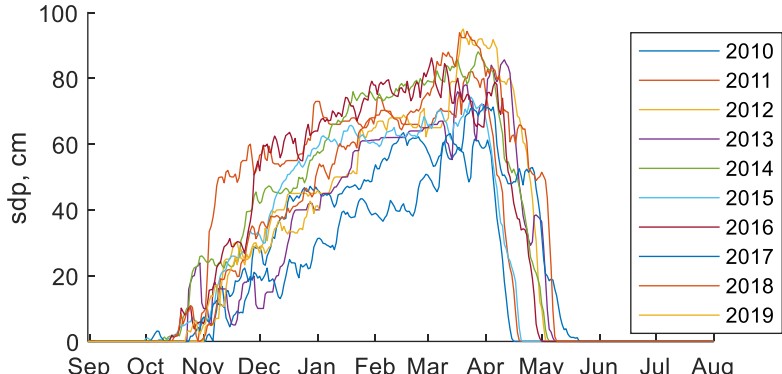

**Figure 5.  Daily snow depth for 2010–2019.**


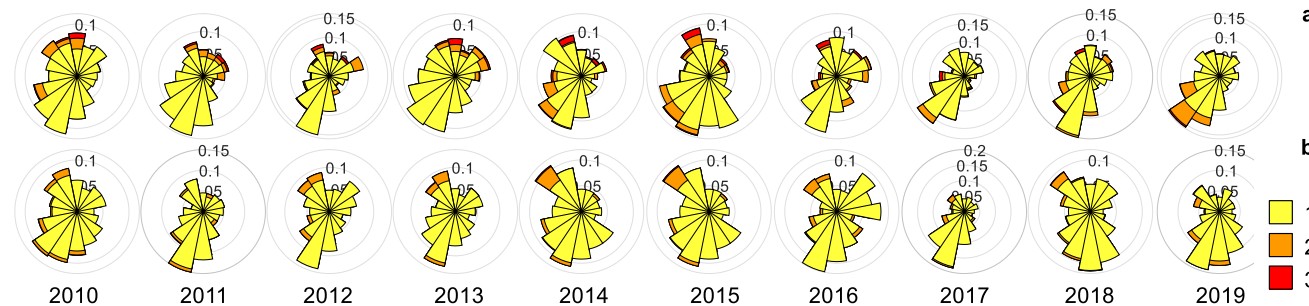

**Figure 6.  Annual wind rose for 2010–2019 at 10 m (a) and 2 m (b). Legend: 1 – wind speed 0.5~2 m s$^{-1}$, 2 – 2~ 5 m s$^{-1}$; 3 - >5 m s$^{-1}$. Wind direction for wind speed below 0.5 m/s was not accounted for plotting.**