# Peer review of "Hydrometeorological dataset of West Siberian boreal peatland: a 10year record from the Mukhrino field station"

_Earth System Science Data, 2020_

## Author Response (AR1)

We thank anonymous Reviewers for evaluating our manuscript. Below we list our responses to each comment. We have updated Figure 1 (and figures at the Suplementary), and have rewritten text according to a specific reviewer's comments.

Reviewer 1 Comments:

o      Line 39: replace "C" with "Carbon"

Reply:

This has been corrected.

o      Line 42: in the citation, it writes "the area of southern-taiga wetlands is estimated at 12.02 Mha at the total wetland area percentage in the subzone estimated at 28%", here in this manuscript, "Large peatland systems in Western Siberia occupy about 28% of the area", are wetland and peatland mean the same thing? Please give a clear description of peatland in the beginning.

Reply:

Peatlands (ecosystems with long-term accumulation of organic matter) represent half of the Earth's wetlands and cover 3% of the global total land area (Wetlands International). West Siberian peatlands are wetlands representing a long-term carbon dioxide sink and global methane source since the early Holocene (Sheng et al., 2004).

o      Line 44: "Only a few modern hydrometeorological datasets are available for the Northern part of Russia", in the citation of Bioke et al., 2019, the research study on a high Arctic permafrost site in the Svalbard archipelago, which is a disputed area.

Reply:

This has been corrected.

o      Line 69: "described in (Lamentowicz et al., 2015)". Please rewrite this part.

Reply:

This has been corrected.

o      Line 71. "described by (Alekseychik et al., 2017)". Please rewrite this part

Reply:

This has been corrected.

o      Line 208: "South-south-east winds prevail at the observation site." But according to Fig. 6, south-south-west winds prevail at the observation site.

Reply:

This has been corrected.

o      Figure 1. Please redraw this figure, which should include a large area with local magnification. Map of peatland distribution is also suggested to include.

Reply:

This has been corrected.

o      In the supplement figures, uniform color in one figure is suggested. Figure S20, in the caption "Top panel: 1-hourly observations using rain gauge, 2-ERA5-Land reanalysis data", and in figure "Mod and Obs", they don't match each other.

Reply:

This has been corrected.